# Functional Mapping of Genes Modulating Plant Shade Avoidance Using Leaf Traits

**DOI:** 10.3390/plants12030608

**Published:** 2023-01-30

**Authors:** Han Zhang, Yige Cao, Zijian Wang, Meixia Ye, Rongling Wu

**Affiliations:** 1Center for Computational Biology, College of Biological Sciences and Technology, Beijing Forestry University, Beijing 100083, China; 2National Engineering Laboratory for Tree Breeding, College of Biological Sciences and Technology, Beijing Forestry University, Beijing 100083, China; 3Key Laboratory of Genetics and Breeding in Forest Trees and Ornamental Plants, Ministry of Education, College of Biological Sciences and Technology, Beijing Forestry University, Beijing 100083, China; 4The Tree and Ornamental Plant Breeding and Biotechnology Laboratory of National Forestry and Grassland Administration, Beijing Forestry University, Beijing 100083, China

**Keywords:** shade avoidance syndrome, allometry, petiole, leaf, functional mapping

## Abstract

Shade avoidance syndrome (SAS) refers to a set of plant responses that increases light capture in dense stands. This process is crucial for plants in natural and agricultural environments as they compete for resources and avoid suboptimal conditions. Although the molecular, biochemical, and physiological mechanisms underlying the SAS response have been extensively studied, the genetic basis of developmental variation in leaves in regard to leaf area, petiole length, and leaf length (i.e., their allometric relationships) remains unresolved. In this study, with the recombinant inbred line (RIL) population, the developmental traits of leaves of *Arabidopsis* were investigated under two growth density conditions (high- and low-density plantings). The observed changes were then reconstructed digitally, and their allometric relationships were modelled. Taking the genome-wide association analysis, the SNP genotype and the dynamic phenotype of the leaf from both densities were combined to explore the allometry QTLs. Under different densities, leaf change phenotype was analyzed from two core ecological scenarios: (i) the allometric change of the leaf area with leaf length, and (ii) the change of the leaf length with petiole length. QTLs modulating these two scenarios were characterized as ‘*leaf shape QTLs*’ and ‘*leaf position QTLs*’. With functional mapping, results showed a total of 30 and 24 significant SNPs for *shapeQTLs* and *positionQTLs*, respectively. By annotation, immune pathway genes, photosensory receptor genes, and phytohormone genes were identified to be involved in the SAS response. Interestingly, genes modulating the immune pathway and salt tolerance, i.e., systemic acquired resistance (SAR) regulatory proteins (MININ-1-related) and salt tolerance homologs (STH), were reported to mediate the SAS response. By dissecting and comparing QTL effects from low- and high-density conditions, our results elucidate the genetic control of leaf formation in the context of the SAS response. The mechanism with leaf development × density interaction can further aid the development of density-tolerant crop varieties for agricultural practices.

## 1. Introduction

Over the course of evolution, plants have acquired an avoidance strategy to adjust their growth when exposed to suboptimal conditions, similar to an escape strategy in animals. Under highly shaded conditions, most plants tailor their photosynthetic strategy to decreased light levels while at the same time expressing shade-avoidance syndrome (SAS). Plants exhibit SAS by positioning leaves into well-lit microsites; accelerating the elongation of hypocotyls, internodes, and petioles; and elongating shoots away from the shade of neighboring plants to reach light. This plays out both in nature and in agricultural fields, for example, in high-density cropping systems. Thus, a detailed understanding of this process could help improve agricultural productivity.

The molecular and physiological underpinnings of the SAS response have been well studied and are covered in previous studies. Chlorophyll in leaf selectively absorb red light (R) and cannot absorb far-red light (FR), so when increasing the plant density, the unabsorbed FR will be reflected by leaves, causing the reduction of the R:FR ratio. To sense the density change, through the detection of FR reflection, the plant inactivates phytochrome (i.e., phyA to phyE). This subsequently modulates a network of transcription factors, hormones, and cell wall-modifying proteins to induce shade avoidance responses [1,2,3]. Upon exposure to R, the cytosolic phyB directly interacts with phytochrome-interacting factors (PIFs) [1]. In particular, PIF4, PIF5, and PIF7 are associated with SAS to induce the hypocotyl elongation [4]. In addition to PIFs, LONG HYPOCOTYL IN FAR-RED1 (HFR1), PHYTOCHROME RAPIDLY REGULATED1 (PAR1), and PAR2 are likely to be the negative regulators of hypocotyl elongation [5,6]. Additionally, in response to low R:FR, ATHB4 are transcriptionally induced to modulate hormone responses [7]. Indole acetic acid (IAA) level in the elongating hypocotyl can specifically be elevated in low R:FR, which appears to be conditional upon PIN3-mediated auxin transport [8]. Additionally, the low ratio of R:FR can induce the expression of GA20ox, which has a function of enhancing the level of GA, triggering the degradation of growth-inhibiting nuclear proteins of DELLA in cascade, and finally bring out an elongated plant phenotype [9]. All this knowledge is essential for promoting the further development of modern agriculture, e.g., increasing crop yields without agricultural expansion, whilst using the density-tolerant genotype under a higher planting density. Although key elements mediating SAS have been identified, we still argue that further work is needed to assess the genetic architecture of SAS, e.g., specific effects of genes in mediating the SAS-inferred phenotypic change, and the mechanistic basis of natural genetic variation in the SAS response.

Recent studies have reported quantitative trait loci (QTLs) that affect the SAS response. In soybean, a major QTL, named *qSAR1*, has been mapped to plant height [10]. In *Arabidopsis*, *EODINDEX1* has been identified as a QTL that strongly impacts hypocotyl inhibition under white light [11]. *ELF3* is a QTL involved in the shade avoidance response index based on leaf morphology traits [12]. Among phenotype changes of the SAS response, the leaf is one of the most important responsive organs, as leaves undergo photosynthesis and detect cue changes in light spectral composition. However, these studies were mainly based on static assessments of leaf traits, whereas leaf formation is a dynamic process that changes during the growth process. In addition, the allometric scaling relationships remain unknown, that is, changes in leaf length and/or shape are very flexible. During ontology, the leaf length varies with petiole length, thus causing the position change of the leaf, which facilitates the leaf to gain more light. These could potentially be modeled based on the allometric relationships between petiole length (P) and leaf length (L), and between leaf length (L) and leaf area (A).

To incorporate the biological scaling allometry law into the mathematical framework of QTL mapping, while retaining the dynamic phenotype of leaf development, we emphasize the superiority of functional mapping [13,14] in locating SAS genes from the aspect of allometry QTLs. Compared with other QTL mapping methods that use the static phenotype from only one timepoint, functional mapping using traits investigated at a series of timepoints can increase the model power for QTL detection, because the developmental process can be modelled well using mathematical equations with functional mapping. For allometry QTLs, through previous studies, genes mediating height-diameter allometry for the tree stem had been computationally identified, with the allometric growth equation equipped in the functional mapping framework [15]. Then, this information can be used with functional mapping to identify the underlying QTL controlling changes in these trait relationships and, thus, deepen our understanding of developmental regulation of SAS in leaves [13,14].

According to the advantage of functional mapping, we design two density conditions and incorporate the dynamic petiole and leaf traits from two density sets into a functional mapping framework. From a methodological viewpoint, this strongly support the detection of SAS QTL, but through a QTL × density interaction approach. Taking an *Arabidopsis* recombinant inbred line (RIL) as the mapping population, we propose the use of bivariate functional mapping to study the genetic architecture of SAS in this study. In particular, how genes modulate the dynamic covariation of petiole with leaf length, how leaf length covaries with leaf area, and how genes mediate the interaction of leaf development and density environment were all comprehensively analyzed in this study.

## 2. Result

### 2.1. Shape Reconstruction

We used the shape reconstruction method described by Fu et al [16]. On principle, with the known size of a black/white square, the coordinate system can be build using the corner point of the checkerboard grid. Figure 1 shows the results for two types of *Arabidopsis* leaf, one with a very short petiole (Figure 1A) and the other with a tenuous petiole (Figure 1B). Using this shape reconstruction method, a series of coordinates along the leaf boundary were determined for both the leaves (Figure 1C,D), which match the photographic images. Figure 1C,D fully possessed the feature of the petiole base from Figure 1A,B. Additionally, the shape with the leaf tip and other parts are reserved, suggesting a strong power with a such method. After the shape reconstruction, the traits of leaf length, leaf area, and petiole length can further be quantified.

### 2.2. Longitudinal Phenotypes of Leaf Traits under Different Density Conditions

Next, using 1680 seedlings, we depicted the phenotypic changes in A, L, and P at the progeny level (Figure 2). Taking all eight timepoints for low-density and seven timepoints for high-density conditions, all three traits showed S-shaped curves during leaf development. In addition, curve-fitting at the population level and between the two conditions, indicated a high variance for all three traits. Overall, as compared with the early stage, the high-density condition led to higher growth rates and larger traits in later stages, supporting the general understanding that the elevation of plant density will cause the alteration of light in quality and quantity, to affect the plant growth or elongation synthetically.

Low-density conditions led to more variance among phenotypes at nearly all timepoints, suggesting an activated SAS response. As shown in Figure 2, by comparing the mean curve from the two-density condition, their crossover points were located at 4.05, 3.67, and 3.23 weeks for the leaf area (Figure 2A), leaf length (Figure 2B), and petiole length (Figure 2C), respectively. The earliest crossover for the petiole trait might imply the sensitivity of petiole growth regulation when facing the changing density condition, suggesting that petiole growth regulation is sensitive to density conditions, and this trait may be the first to trigger the process of SAS.

### 2.3. Mathematical Modeling of Leaf Allometry

To verify the fitness between the allometric growth equation and the measured phenotypes, the R-squared value for plants grown under each condition was evaluated using the least squares method. Under high-density conditions, more than 71% of progenies had R-squared values greater than 0.73 for the P-A relationship. For the L-A and P-L relationships, more than 93% and 92% of progenies had R-squared values greater than 0.96 and 0.92, respectively. Under low-density conditions, 75%, 92%, and 90% of progenies had goodness-of-fit values of 0.75, 0.94, and 0.95, respectively. These results indicate that the allometry between P and A was not strong. Therefore, we subsequently focused on analyzing allometry for the latter two pairs of traits, i.e., L-A and P-L.

Figure 3 shows the allometric patterns for L-A and P-L during development. L-A has a concave-shaped curve (Figure 3A,B), whereas that for P-L is convex (Figure 3D,E). Although the allometric pattern for each progeny varied, those associated with each density were remarkably similar. The data in Figure 3C indicate that, for a given lamina area, high-density conditions led to slightly longer leaves, indicating that leaf elongation occurs to ensure light acquisition in such environments. Nevertheless, petiole lengths were nearly equal between the two densities (Figure 3F). This suggests that, under high density, petiole elongation does not necessarily account for the increase in leaf length. This coincides with the nonsignificant goodness-of-fit of the allometry data for P-A. Considering the substantial role of the petiole in changing the leaf position and the considerable variation in P-L under both densities, the allometric relationships between P and L can provide important insights into the genetic dissection of the SAS mechanism. By estimating the power coefficients from Equation (1), how one trait scales with the other can be illustrated, and the scaling pattern between the two densities can be compared.

### 2.4. QTLs for SAS

By integrating phenotypes from the high- and low-density groups into the QTL mapping framework, a bivariate functional mapping model incorporating Equation (1) was implemented. We tested how QTL regulates the allometric scaling of L-A and P-L during leaf ontogeny. We argue that L-A and P-L allometry has different developmental significance for leaves. For L-A, at any given length, leaves with a smaller A are narrower, and this might lead to better photosynthetic ability than larger leaves, increasing competitiveness under crowded and shaded conditions.

To help the interpretation of QTLs, we performed a gene enrichment analysis using the combined genes of two set of analysis. The enrichment result based on Gene Ontology (GO) showed six classes of functions with *p-*values less than 0.05: red, far-red light phototransduction (GO:0009585, 2.56 × 10^−7^), gibberellin 20-oxidase activity (GO:0045544, 3.28 × 10^−5^), shade avoidance (GO:0009641, 0.0014), response to light stimulus (GO:0009416, 0.038), leaf shaping (GO:0010358, 0.021), regulation of defense response (GO:0031347, 0.015). In comparison with the background gene number, 8/32, 4/8, 4/26, 2/12, 2/28, 5/83 genes were involved in these functions, respectively. In particular, for leaf development, the detailed annotation of ‘leaf pavement cell development’, ‘leaf shaping’, and ‘leaf vascular tissue pattern formation’ were found. For class 6, the ‘defense response to bacterium’, ‘regulation of systemic acquired resistance’, ‘defense response to fungus’, ’response to UV-B’, and ‘defense response to bacterium, incompatible interaction’ were found. All these have pointed out the importance of leaf development, immune system, far-red light regulation to the SAS, and have also promoted the screening of key genes for the further dissection of SAS.

### 2.5. ShapeQTLs: How Leaf Area Scales with Leaf Length

Supporting the general understanding that the elevation of plant density will cause the alteration of light in quality and quantity, this will affect the plant growth or elongation synthetically. Of these, 280, 82, 86, 90, and 7 were distributed between chromosomes 1 to 5 (Figure 4A). All were annotated using the TAIR website (www.arabidopsis.org, accessed on 1 October 2022) according to their genomic position, resulting in 30 annotated functional proteins. In terms of protein function, there were fourteen photosensory receptor genes, four phytohormone genes, six immune pathway genes, and one leaf shape-related gene (Table 1), which implies that SAS is regulated through a multi-step mechanism. The photosensory receptor pathway had the greatest number of genes, suggesting its paramount role in regulating the SAS response. For these key genes, their phenotype variation explained (PVE) was calculated, ranging from 0.99% to 1.91%.

Next, we focused on an SNP located at AT4G01895.1, which encodes systemic acquired resistance (SAR) regulatory proteins (NIMIN-1-related) belonging to the immune regulation pathway. As shown in Figure 5A, each genotype at this shapeQTL showed a difference in the pattern of how leaf area and leaf length change with time. With the two-dimensional expansion of the leaf, A showed a period of linear growth from 2 to 4 weeks and increased substantially, compared to the increase in L. Indeed, the curves for A showed greater differences between density conditions than between genotypes. Over time, the allometric difference between the two densities narrowed (Figure 5B). Regarding genotypes, *CC* and *GG* showed significant differences between the two densities. In addition, all 10 *shapeQTLs* showed a weakening impact on the L-A relationship during the middle-late stages of growth after a slight increase in the early stage (Figure 6A). Once a leaf reaches 4 cm in length, the additive effects decrease. In comparing effects between the two densities, the inversive patterns of 10 additive effect curves were disclosed (Figure 6B).

### 2.6. PositionQTLs: How Leaf Length Scales with Petiole Length

There were 418 significant SNPs identified for P-L allometry, involving 280, 30, 17, 59, and 32 SNPs on chromosomes 1 to 5 (Figure 4B). Functional annotation identified 24 protein genes classified into the categories listed in Table 2. The pathways and gene categories were the same as those for *shapeQTLs*, clearly indicating their shared function in regulating SAS. Therefore, *positionQTLs* not only regulate leaf position by changing L or P, but also affect traits that are regulated by *shapeQTLs*. These key genes owned PVEs ranging from 1.01% to 1.88%.

Next, we focused on a QTL located at AT3G22170.1, which encodes far-red elongated hypocotyls 3 protein. Under both densities, P changed with time; although its growth rate was slower than that of L, both P and L changed more at the density level than at the genotype level (Figure 5C). High-density conditions led to a larger P than low-density conditions, which suggests that this QTL plays a role in developing a longer petiole to move leaves towards well-lit microsites. For L, the curves diverged between genotypes after 1.5 weeks under low density. Using the estimated parameters from Equation (1), we analyzed the allometry pattern at the genotype level. The *positionQTLs* fit the allometry scaling relationship for P-L well (Figure 5D). The allometric patterns among QTL genotypes under high density were indistinguishable. This suggests a G × E interaction property with this QTL. For genotypes *AA* and *TT*, the differences in allometry scaling between the two densities were significant. Calculations of the additive effects under low density indicated that QTLs 4, 5, and 7 showed a downward parabola-shaped additive effect. QTLs 3, 6, 9, and 10 showed a sharp decrease with an increase in P (Figure 6C). Under high density, QTLs 1, 2, 3, 6, 9, and 10 showed a decreasing role in controlling allometry in relation to L. QTLs 5, 7, 8, and 9 showed an increasing role with P (Figure 6D).

**Figure 6 plants-12-00608-f006:**
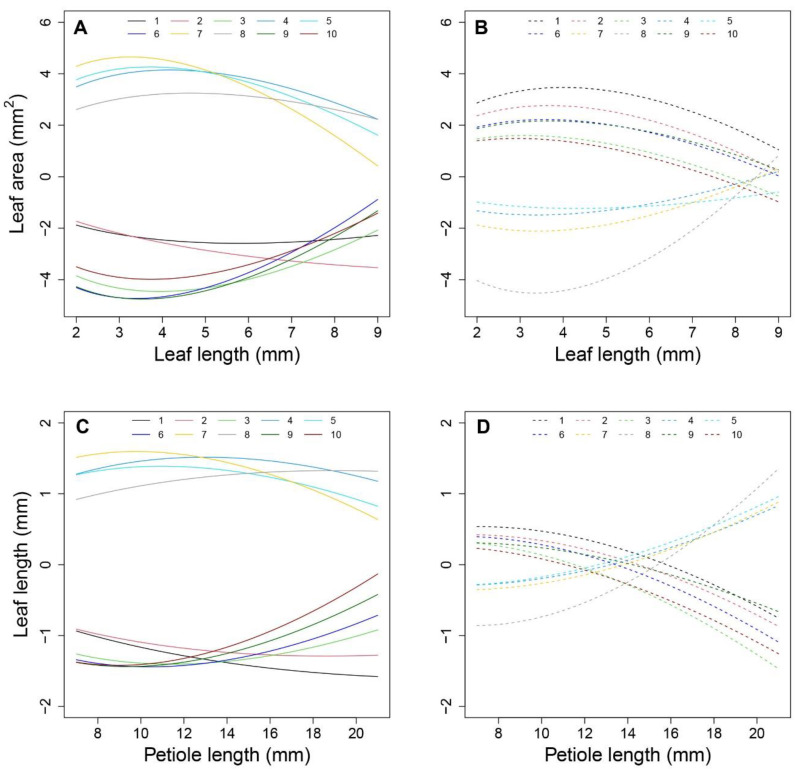
Additive effect with significant SNPs. (**A**,**B**) show the additive effect of *shapeQTLs* with 10 significant genes under low- and high-density conditions. Numbers 1–10 present genes AT1G17020.1, AT1G19850.1, AT2G03340.1, AT2G27110.1, AT3G59060.1, AT3G63300.1, AT4G16250.1, AT4G25420.1, AT5G28530.1 and AT5G40910.1, respectively. (**C**,**D**) show the additive effect of *positionQTLs* with 10 significant genes under low- and high-density conditions. Numbers 1-10 present AT1G29690.1, AT1G10240.1, AT1G51090.1, AT2G43010.1, AT3G22170.1, AT3G58850.1, AT4G18130.1, AT4G27430.1, AT5G38860.1 and AT5G51810.1, respectively.

### 2.7. Pleiotropism among QTLs

Interestingly, 89 significant SNPs were identified for *shapeQTLs* that were also significant SNPs for *positionQTLs*, which suggests that these SNPs simultaneously govern allometric scaling of both L-A and P-L. There were 55, 6, 5, 21, and 2 shared SNPs located on chromosomes 1 to 5. As the two allometric relationships have different biological meanings, these QTLs appear to be pleiotropic. AT1G10240.1, AT1G69010.1, and AT5G38860.1 were directly associated with the SAS response. These SNPs encode FAR1-related sequence 11, BES1-interacting Myc-like protein 2, and BES1-interacting Myc-like protein, respectively, and were all involved in photosensory pathways. The QTL at AT1G10240.1 had a significant role in regulating L-A (Figure 7A,B) and P-L allometry under both densities (Figure 7C,D). Such a QTL may behave more sensitively to alter the phenotype of the leaf when density changes, so that plants can produce a flexible mechanism to better adapt to shade.

## 3. Discussion

SAS triggers phenotypic changes in plants, as they attempt to avoid shade and seek enhanced light interception. For plants, shade varies both in the vertical and horizontal directions and also temporally, and shade avoidance can involve any one or more of these axes. All these hold important implications for agriculture practice. For example, in the context of decreasingly available arable land and the increasing population, rational close cultivation is one of the effective strategies to increase crop yield per unit area. To date, although studies have identified dominant components of SAS regulation, the dissection on genetic control of SAS plasticity had largely not factored the co-varying relationships among distinct phenotypes of leaf.

In this study, we focused on the allometric relationships among leaf traits. Specific traits were investigated longitudinally and nondestructively by modeling the changes observed in laboratory experiments. We also modeled the allometric relationships among traits and determined how QTLs may control allometry. The association analysis for each SNP have been used to determine the genetic architecture of complex traits [17,18]. Functional mapping, a QTL identification framework that utilizes mathematical curves to model biological growth, was conceptualized two decades ago [13]. Although only one mapping population used in this study, we hope to construct a second mapping population to validate the QTL results to improve the reliability. Compared to GWAS using static traits, the use of dynamic phenotypes can provide deeper biological insights by correlating genotypes with phenotypes. Thus, since this method was introduced, it has been extensively used, for example, to dissect gravitropism and phototropism based on dynamic phenotypes of seedlings [19], to study leaf allometry between leaf area and leaf mass in the common bean [20], and to model phenotypic plasticity using growth trajectories of rice [21].

In this study, L-A and P-L relationships were used as a predictor of the capacity of *Arabidopsis* to respond to shade. Through functional mapping, the genetic architecture underlying allometry was dissected. Focusing on traits of leaf development and incorporating phenotypes from plants grown under two different densities, we were able to map *shapeQTLs* and *positionQTLs*. We also identified pleiotropic genes. Among genes modulating the change in A as a function of L, we identified 545 functional genes in the *shapeQTLs* group, 30 of which encode a molecular function: these included photosensory receptor genes, phytohormone genes, and genes regulating leaf development and leaf shape. Most photosensory receptors were phytochromes, which respond to low R:FR ratios caused by dense vegetation [22]. AT3G63300.1, encoding FORKED 1 (FKD1), plays a role in patterning leaf vascular and can coordinate leaf size with vein density [23]. Interestingly, AT1G75540.1, which encodes salt tolerance homolog2 (STH2), was also associated with shade avoidance. It interacts with COP1 to control de-etiolation and positively regulates photosynthesis [24]. Interestingly, high-density growth conditions lead to more severe plant diseases [25] which may be due to the change in light environment (e.g., a decrease in UV-B radiation typically decreases plant resistance to biotic stressors).

Genes modulating changes in L as a function of P (i.e., *positionQTLs*) fell into the same pathway categories as those for *shapeQTLs* but the gene sets were different. This implies that changes in leaf position and leaf shape are linked to the SAS response. The overlap of gene categories between L-A and P-L may reflect their combined role in driving changes in leaf position and shape. On the other hand, this overlapping further illustrated the need for QTL mapping of SAS using distinct allometric trait pairs; indeed, doing so identified another phenotype regulating SAS under high-density conditions, namely, the A trait. This indicates that there are QTL–density interactions. Remarkably, a previous study reported the role of AT3G22170 in mediating the elongation of hypocotyls in response to FR light in concert with FAR1 [26,27]. However, in our study, this QTL was related with P-L, revealing another role during later development.

The allometry underlying complex traits is pervasive. For example, the metabolic theory of ecology explaining metabolic rate scales with the three-fourths power of body mass [28]. Another example is the quarter-power scaling relationship between separate organs and overall body size [29]. As a practical case, in forms of height-diameter allometry [15] identified genes modulating the tree stem growth. Using Reeve and Huxley’s allometric equation [30], the increase in the dry weight of soybean seed was examined using a functional mapping method [31]. Therefore, the allometry principle deserves a widespread application to the dissection of more complex traits. In this study, we used functional mapping to map allometry-related QTLs, with an emphasis on SAS. We utilized an RIL population, which can give sufficient replicate samples, as each progeny within the RIL population can propagate many replicates through self-reproduction. This can reduce spurious associations when mapping QTLs. In addition, we reduce other factors by growing plants in chambers with controllable conditions. All these steps improved the precision of QTL detection.

For rosette species such as *Arabidopsis*, vegetative internode elongation is highly suppressed and we, therefore, used leaf traits to analyze the genetic architecture of SAS. However, for other non-rosette plants, accelerated elongation of internodes and the resulting increase in stem length have been reported as the most striking SAS component [32]. This might also be accompanied by enhanced apical dominance and stimulated vertical growth of the main stem. In such species, it may be better to adjust the model to consider the growth relationship between internode length and stem height. The further modeling of such potential allometry could reveal molecular mechanisms underlying dwarf or semidwarf phenotypes, which could aid the development of modern crop species that can grow at high densities.

## 4. Materials and Methods 

### 4.1. Plant Materials 

*Arabidopsis* ecotype *Ler* (*Landsbergerecta*) and *Sha* (*Shakdara*) were used as parental material. F1 seeds were obtained by manual crossing and subsequent self-fertilization for 10 generations to develop a RIL population. In all, 84 progenies were used, with each progeny, 20 replicates were taken for both the high- and low-density conditions. The genome DNA for the whole RIL population was extracted with a TIANGEN DP305 DNA extraction kit, using 100 mg of young leaf. DNA quality was inspected by agarose electrophoresis and imaged with an ultraviolet imager. DNA samples without degradation were sequenced subsequently. The RIL population was genotyped at Total Genomics Solution (TGS) Institute, China. Using the Illumina Hiseq 2000 platform, the 125 bp paired-end reads were produced at a depth of 9.91×. After filtering the low-quality sequence, 1.10 GB data per sample were generated on average. The percentage of Q30 and Q20 bases were more than 90% and 95%, respectively, with a normal GC distribution. Taking the reference genome from the TAIR website (https://www.arabidopsis.org/download/index-auto.jsp?dir=%2Fdownload_files%2FSequences%2FAssemblies, accessed on 1 October 2022), SNP detection was performed with GATK, a widely used mutation detection software. The SNP screening criteria were customized as follows: the sequencing depth ≥ 4, the SNP base quality ≥ 20, the variation detection quality ≥ 50. SNPs with the allele frequency ≥ 5%, or with the missing rate ≥ 50% were filtered out. With these criteria, a total of 417,495 high-quality SNPs were determined for QTL mapping [33].

### 4.2. Experimental Design

From the 84 progenies, 40 seeds were sowed, from which 20 seedlings were used for high- and low-density planting (with 3 × 3 cm and 5 × 5 cm row spacing for the high and low conditions, respectively). For each density condition, a randomized block design of 20 replicates with each progeny, 5 blocks each with 4 replicates, was used. Plants were grown in an artificial climate chamber at Beijing Forestry University, China. Light and temperature conditions in the chamber were 16 h light and 8 h dark per day, with a light intensity of 150 μmol·s^–1^·m^–2^ and a constant room temperature of 22 °C with 80% relative humidity. A mixture of turfy soil, perlite, and vermiculite at a 1:1:1 ratio was used for cultivation. 

### 4.3. Phenotype Collection 

One week after transplanting, the third true leaf of each plant was chosen for phenotypic investigation. Selected leaves were photographed weekly using a digital camera (NIKON COOLPIX A10). To assess phenotypes, an image calibration card (a printed checkerboard, each square having a size of 10 × 10 mm) was used as illustrated in Figure 1. At the time of taking the photograph, the leaf was flattened against the blank space of the card, making the checkerboard and leaf integrity show on photo (Figure 1A,B). The information was imported into the CameraCalibrator module of MATLAB for further analysis. Then, following a previously described method [16], all leaf shapes were reconstructed. Furthermore, the three traits under study (P, L, and A) were assessed via MATLAB. 

### 4.4. Statistical Models 

Mean phenotypes were calculated for each progeny in the RIL population, and each of the three traits were calculated. Equation (1) was used to model allometric growth:(1)$$\mu_{j}=\alpha_{j}{\left(x_{i}\right)}^{\beta_{j}}$$
where *x_i_* is the longitudinal phenotype of one trait for progeny *i*, *μ_j_* is the mean vector of another trait for SNP genotype *j*, and *α_j_* and *β_j_* are parameters for the allometric law. 

To incorporate the dynamic phenotype from the two densities into one model, taking the scaling relationship of L-A as an example, let ***x****_i_* = (***x****_ih_*; ***x****_il_*) = (***x****_ih_*(*t*_1_),(***x****_ih_*(*t*_2_),…, (***x****_ih_*(*t*_Th_); ***x****_il_*(*t*_1_),(***x****_il_*(*t*_2_),…, (***x****_il_*(*t*_Tl_)) denote the joint phenotype of L for progeny *i*, where *h* represents the high-density environment and *l* represents the low-density environment. Similarly, ***y****_i_* = (***y****_ih_*; ***y****_il_*) = (***y****_ih_*(*t*_1_),(***y****_ih_*(*t*_2_),…, (***y****_ih_*(*t*_Th_); ***y****_il_*(*t*_1_),(***y****_il_*(*t*_2_),…, (***y****_il_*(*t*_Tl_)) expresses the joint phenotype of A. For the two-density condition, we allowed the starting measurement time to vary depending on the growth after seedling transplantation.

Next, assuming two genotypes for each SNP, *n* progenies can be grouped into two sets. The likelihood for the dynamic trait from the two-density condition is formulated as follows:(2)$$L\left(y\right)=\overset{n}{\underset{i=1}{\prod }}f_{j|i}\left(\left(y_{ih};\;y_{il}\right)\;|\mu_{j},\Sigma_{i}\;\right)$$

According to this equation, at a specific SNP, the given progeny *i* carrying QTL genotype *j*, the function $f_{j|i}\left(\left(y_{ih};\;y_{il}\right)\;|\mu_{j},\Sigma_{i}\;\right)$ is a multivariate normal distribution, where the expected mean vector *μ_j_* is expressed as
(3)$$\mu_{j}=(\mu_{jh};\;\mu_{jl})=(\mu_{jh}\left(t_{1}\right),\mu_{jh}\left(t_{2}\right),\ldots ,\mu_{jh}\left(t_{Th}\right);\;\mu_{jl}(t_{1}),\;\mu_{jl}(t_{2}),\;\mu_{jl}(t_{Tl})$$

For (co-)variance matrix *∑_i_*, because *x_i_* in Equation (1) denotes one trait within the allometric trait pair, *x_i_* has progeny-specific values. To synthesize the two density conditions, *∑_i_* can be expressed as
(4)$$\Sigma_{i}=\left(\begin{array}{cc} \Sigma_{i\left(hh\right)} & \Sigma_{i\left(hl\right)}\\ \Sigma_{i\left(lh\right)} & \Sigma_{i\left(ll\right)} \end{array}\right)$$
where *∑_i(hh)_* and *∑_i(ll)_* represent a (*T_h_* × *T_h_*) and (*T_l_* × *T_l_*) residual covariance matrix of growth over the *y* trait, respectively, and *∑_i(hl)_* = (*∑_i(lh)_*)’ is a (*T_h_* × *T_l_*) covariance matrix between the two density conditions. Based on a previous derivation of multi-trait functional mapping [34], combined with a one-order structured ante-dependence model (SAD (1)) [35] we further rewrite (4) as
(5)$$\Sigma_{i}=\Lambda \Sigma_{\varepsilon }\Lambda^{\prime }$$
where
(6)$$\Lambda =\left(\begin{array}{ccccc} I & 0 & 0 & \cdots & 0\\ V^{\left(T_{2}–T_{1}\right)} & I & 0 & \cdots & 0\\ V^{\left(T_{3}–T_{1}\right)} & V^{\left(T_{3}–T_{2}\right)} & I & \cdots & 0\\ \vdots & \vdots & \vdots & \ddots & \vdots \\ V^{\left(T_{T}–T_{1}\right)} & V^{\left(T_{T}–T_{2}\right)} & V^{\left(T_{T}–T_{T–1}\right)} & \cdots & I \end{array}\right)$$
and $V=\left(\begin{array}{cc} \varphi_{1} & 0\\ 0 & \varphi_{2} \end{array}\right)$; $\;\varphi_{1}$ and $\varphi_{2\;}$ refer to the parameters of the two-dimensional SAD (1), $\Sigma_{\varepsilon }$= diag($\sigma^{2}$).

A maximum likelihood estimation was used to estimate curve parameters ($\alpha_{j}$,$\beta_{j}$) = ($\alpha_{jh}$,$\alpha_{jl}$,$\beta_{jh}$,$\beta_{jl}$) for the mean vector, and (co-)variance parameters ($\varphi_{1}$, $\varphi_{2\;}$, $\sigma^{2}$). Under likelihood (2), whether a QTL for SAS regulation exists can be tested by formulating the null and alternative hypotheses as follows:
(7)$$H_{0}:\;\alpha_{j}=\alpha ,\;\beta_{j}=\beta \;H_{1}:\;\mathrm{At}\;\mathrm{least}\;\mathrm{one}\;\mathrm{of}\;\mathrm{the}\;\mathrm{above}\;\mathrm{expression}\;\mathrm{does}\;\mathrm{not}\;\mathrm{hold}$$

For this hypothesis, we used the likelihood ratio test to determine QTLs, the threshold for which can be determined through 1000 replicates of a permutation test, using a significance level of 0.01.
(8)$$LR=–2\left(logL_{0}–logL_{1}\right)$$

We can also test how QTLs may change allometric scaling relationships under different densities to impact the SAS response. This can give deeper insights into the genetic control of leaf traits to optimize photosynthetic properties. The equation can be expressed as follows:
(9)$$H_{0}:\;\alpha_{jh}=\alpha_{jl},\;\beta_{jh}=\beta_{jl}\;H_{1}:\;\mathrm{At}\;\mathrm{least}\;\mathrm{one}\;\mathrm{of}\;\mathrm{the}\;\mathrm{above}\;\mathrm{expressions}\;\mathrm{does}\;\mathrm{not}\;\mathrm{hold}$$

We performed the same calculations for P and L to determine their allometric relationship. Then, considering the three correlated leaf traits (P, A, and L), the two allometric scaling relationships, and the two (high/low) density conditions, the QTLs responsible for the SAS response were comprehensively assessed.

### 4.5. Functional Annotation

Using the models above, all SNPs with a high LOD value larger than the threshold were taken as QTLs. All SNPs were aligned against the Arabidopsis genome; thus, the annotation result with all SNPs can be determined, including their genomic position, and their reported function if SNPs were located at functional proteins.

## 5. Conclusions

Using the developmental trait of leaf, this study performed the QTL mapping of plant SAS with the Arabidopsis RIL population. With the aid of bivariate functional mapping, leaf traits investigated under a high- and low-density planting environment were mathematically modeled from the viewpoint of allometry. For *shape*QTL and *position*QTL results using the L-A and P-L allometry, a total of 30 and 24 functional SNPs were determined to mediate the plant SAS, respectively. Through functional annotation, immune pathway genes, photosensory receptor genes, and phytohormone genes were identified to be involved in the SAS response, among which, systemic acquired resistance (SAR) regulatory proteins (NIMIN-1-related) and salt tolerance homolog2 (STH2) were identified. By dissecting and comparing QTL effects, the results elucidate the genetic control of leaf formation in the context of the SAS response.

## Figures and Tables

**Figure 1 plants-12-00608-f001:**
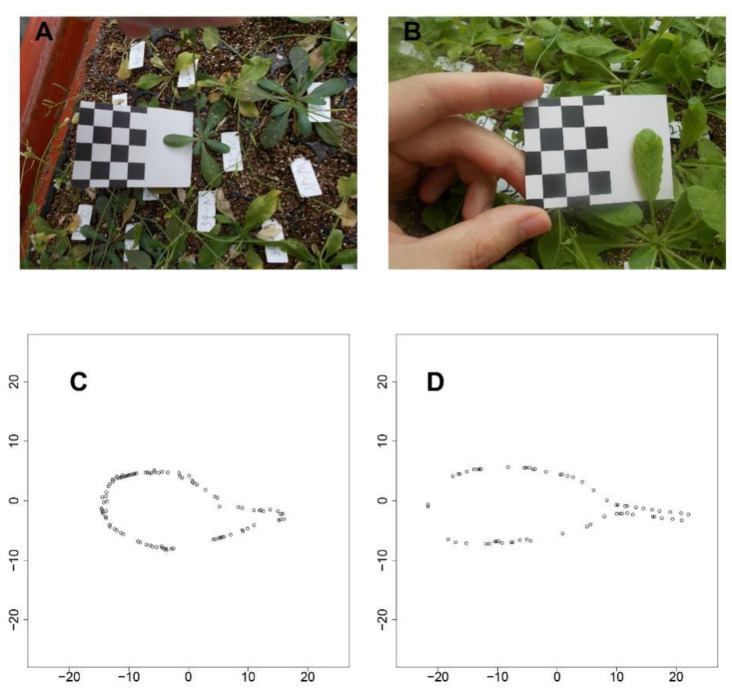
Effect with the shape reconstruction procedure. (**A**,**B**) are the original images taken by a digital camera, (**C**,**D**) are the reconstructed shapes manifested in coordinate systems.

**Figure 2 plants-12-00608-f002:**
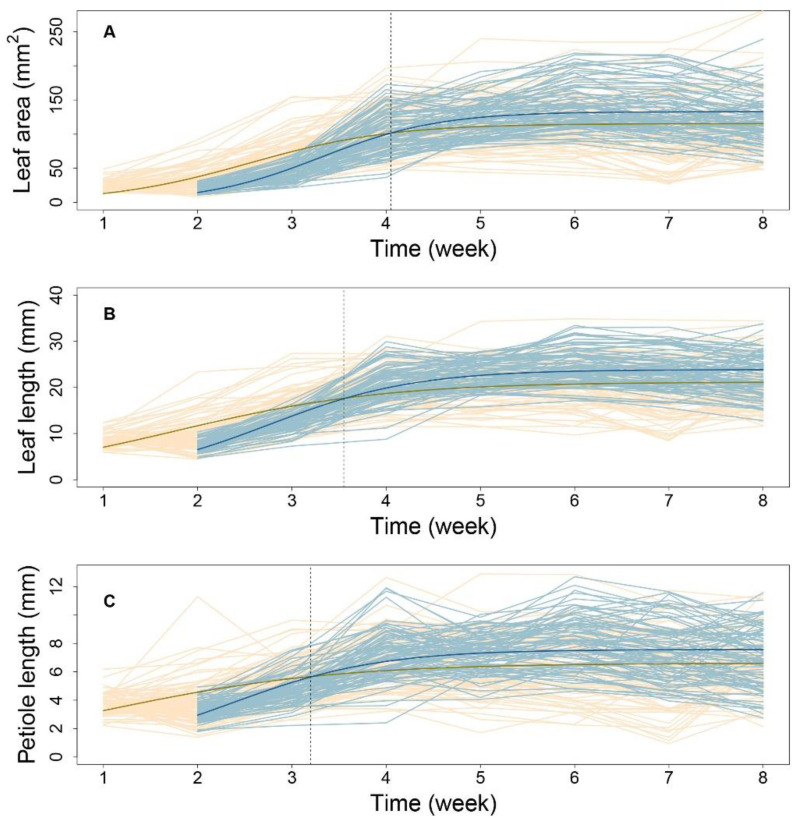
Dynamic growth pattern of leaf traits under high-density and low-density conditions. (**A**–**C**), respectively, present the ontogenic growth curve of the leaf area, leaf length, and petiole length. Yellow curves denote trait patterns from a low-density environment while blue curves denote that from a high-density environment. The two thick lines represent the fitted mean growth curve, while thin lines represents the observed growth phenotype with each progeny in the population. The vertical line indicates the timepoint when the two mean curves for the two densities crossed.

**Figure 3 plants-12-00608-f003:**
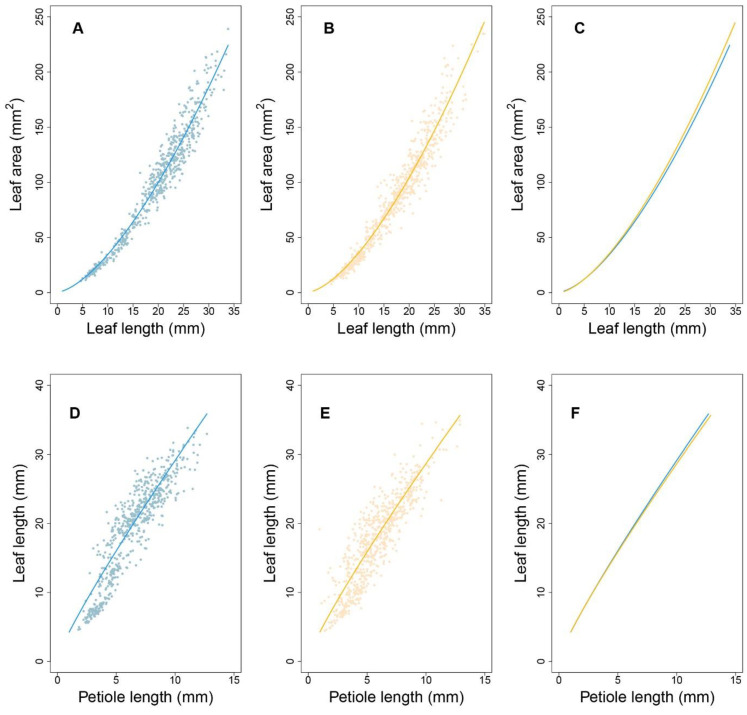
Fitness of allometric growth with different traits of leaf. (**A**,**B**) are the allometric pattern between the leaf length and leaf area, respectively, for high- and low-density conditions. (**D**,**E**) are the pattern between the petiole length and leaf length for high- and low- density conditions. (**C**,**F**) are the comparison of the mean allometric curve fitted for leaf length–leaf area and petiole length–leaf length, respectively.

**Figure 4 plants-12-00608-f004:**
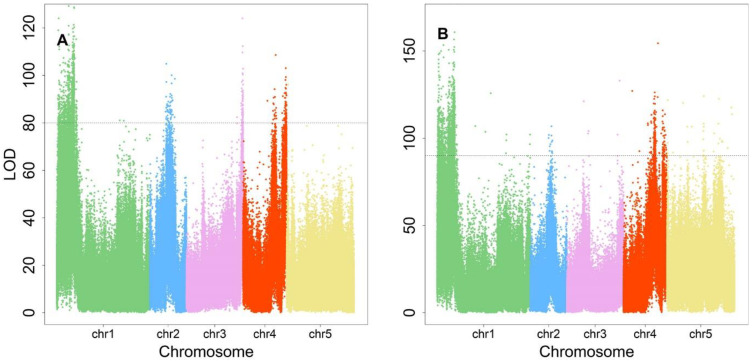
Manhattan plot of association results with shade avoidance syndrome. (**A**,**B**) are association results for the allometry association results for leaf length–leaf area and petiole length–leaf length, respectively. The dashed horizontal line represents the LOD threshold at the significance level of 0.05.

**Figure 5 plants-12-00608-f005:**
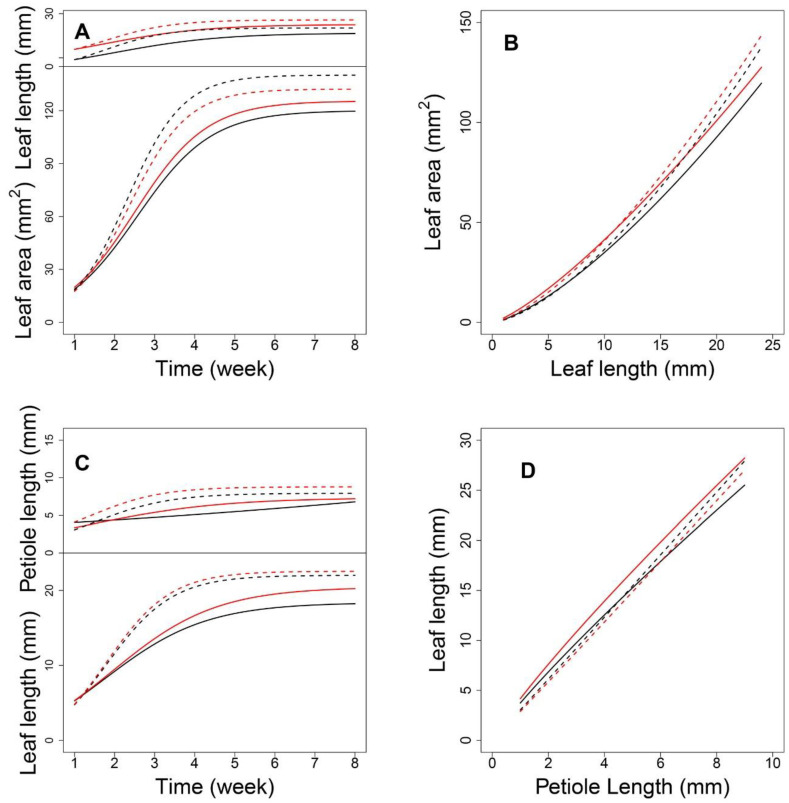
Expression patterns of allometry QTLs. With QTL located at AT4G01895.1, (**A**) shows the ontogenic growth curve of leaf length and leaf area, and (**B**) shows their allometry growth curve. With QTL located at AT3G22170.1, (**C**) shows the ontogenic growth curve of petiole length and leaf length, and (**D**) shows their allometry growth curve. The solid line represents low density, the dotted line represents high density, and red and black color express different genotypes with the QTL.

**Figure 7 plants-12-00608-f007:**
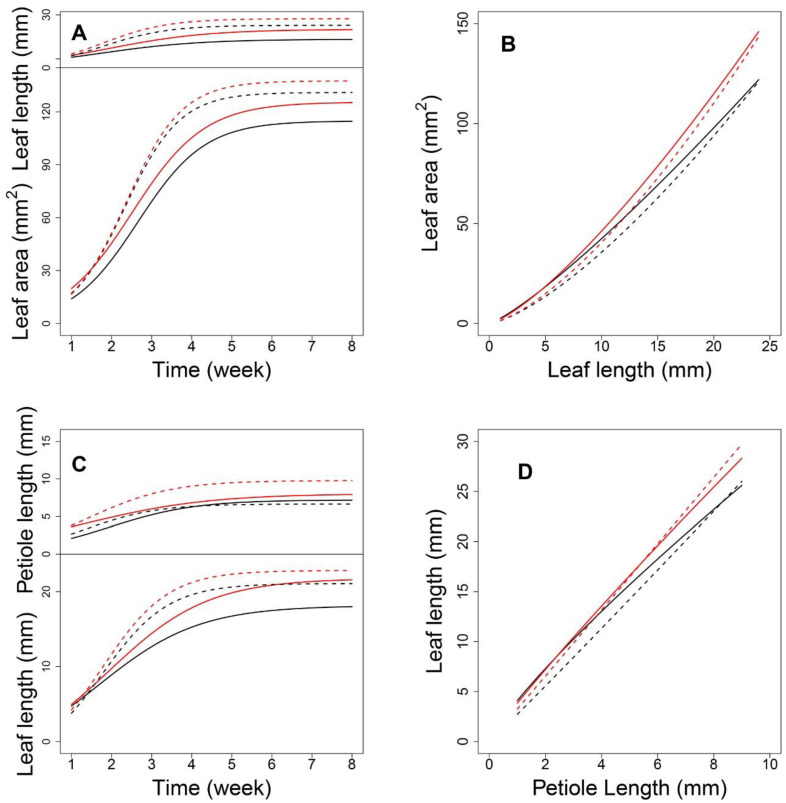
Expression patterns of pleiotropic QTL. With QTL located at AT1G10240.1, (**A**) shows the ontogenic growth curve of leaf length and leaf area, and (**B**) shows their allometry growth curve. (**C**) shows the ontogenic growth curve of petiole length and leaf length, and (**D**) shows their allometry growth curve. The solid line represents low density, the dotted line represents high density, and red and black color express different genotypes with the QTL.

**Table 1 plants-12-00608-t001:** Functional annotation of QTLs associated with allometry between leaf length and leaf area.

Type	Chromosome	Location	Gene ID	Gene Description	PVE
Immune pathway	1	5821947	AT1G17020.1	Senescence-related gene 1 (SRG1)	1.312%
	1	11605859	AT1G32210.1	Death resistance protein	1.415%
	2	1021553	AT2G03340.1	WRKY DNA-binding protein (WRKY33)	1.502%
	2	7308099	AT2G16870.1	TIR-NBS-LRR-like resistance protein	1.004%
	4	818434	AT4G01895.1	Systemic acquired resistance (SAR) regulatory proteins NIMIN-1-related	1.628%
	5	16392619	AT5G40910.1	TIR-NBS-LRR-like resistance protein	1.012%
Photosensory receptor genes	1	3359050	AT1G10240.1	FAR1-related sequence 11 (FRS11)	1.911%
	1	19567276	AT1G52520.1	FAR1-related sequence 6 (FRS6)	1.328%
	2	11581554	AT2G27110.1	FAR1-related sequence 3 (FRS3)	1.401%
	5	10525858	AT5G28530.1	FAR1-related sequence 10 (FRS10)	1.408%
	1	3100411	AT1G09570.1	Phytochrome A	1.909%
	2	8144368	AT2G18790.1	Phytochrome B	1.650%
	4	9203572	AT4G16250.1	Phytochrome D	1.341%
	2	8708022	AT2G20180.1	Phytochrome interacting factor 3-like 5 (PIL5)	1.406%
	3	21829916	AT3G59060.1	Phytochrome interacting factor 3-like 6 (PIL6)	1.326%
	1	25946396	AT1G69010.1	BES1-interacting Myc-like protein 2 (BIM2)	1.073%
	5	15558746	AT5G38860.1	BES1-interacting Myc-like protein 3 (BIM1)	0.991%
	2	17837651	AT2G42870.1	PHY rapidly regulated 1 (PAR1)	1.212%
	2	344412	AT2G01800.1	COP1-interacting protein-related (CIP)	1.303%
	5	14727698	AT5G37190.1	COP1-interacting protein 4 (CIP4)	1.531%
Phytohormones	1	2218953	AT1G07240.1	UDP-glucosyl transferase 71C5	1.307%
	1	25051840	AT1G67080.1	Abscisic acid (aba)-deficient 4	1.721%
	4	12993626	AT4G25420.1	2-Oxoglutarate (2OG) and Fe(II)-dependent oxygenase superfamily protein	1.405%
	4	13221364	AT4G26080.1	Protein phosphatase 2C family protein (PP2C)	1.702%
Leaf development-related	1	1245253	AT1G04550.1	AUX/IAA transcriptional regulator family protein (AXR3)	1.356%
	1	5883293	AT1G17220.1	Translation initiation factor 2, small GTP-binding protein (FUG1)	1.377%
	1	6152210	AT1G17870.1	Ethylene-dependent gravitropism-deficient and yellow-green-like 3 (EGY3)	1.398%
	1	6882081	AT1G19850.1	Transcriptional factor B3 family protein/auxin-responsive factor AUX/IAA-related (MP)	1.395%
Leaf morphology correlation	3	23386503	AT3G63300.1	FORKED 1 (FKD1)	1.281%
Others	1	28362736	AT1G75540.1	Salt tolerance homolog2 (BBX21)	1.341%

**Table 2 plants-12-00608-t002:** Functional annotation of QTLs associated with allometry between the petiole length and leaf length.

Type	Chromosome	Location	Gene ID	Gene Description	PVE
Immune pathway	1	10379838	AT1G29690.1	MAC/Perforin domain-containing protein (NSL1)	1.310%
	1	18933188	AT1G51090.1	Heavy metal transport/detoxification superfamily protein (NAKR1)	1.202%
	1	24067759	AT1G64790.1	ILITYHIA (ILA)	1.344%
Photosensory receptor genes	1	3359050	AT1G10240.1	FAR1-related sequence 11 (FRS11)	1.011%
	4	10832385	AT4G19990.1	FAR1-related sequence 1 (FRS1)	1.142%
	4	17905395	AT4G38170.1	FAR1-related sequence 9 (FRS9)	1.149%
	1	25946396	AT1G69010.1	BES1-interacting Myc-like protein 2 (BIM2)	1.261%
	5	15558746	AT5G38860.1	BES1-interacting Myc-like protein 3 (BIM3)	1.447%
	2	17881666	AT2G43010.1	Phytochrome interacting factor 4 (PIF4)	1.282%
	3	7826998	AT3G22170.1	Far-red elongated hypocotyls 3 (FHY3)	1.889%
	3	21759608	AT3G58850.1	PHY rapidly regulated 2 (PAR2)	1.156%
	4	13723181	AT4G27430.1	COP1-interacting protein 7 (CIP7)	1.600%
	4	10049797	AT4G18130.1	Phytochrome E	1.717%
	5	14015733	AT5G35840.1	Phytochrome C	1.456%
Phytohormones	1	12448321	AT1G34170.1	Auxin response factor 13	1.335%
	1	21976267	AT1G59750.1	Auxin response factor 1	1.112%
	1	27657184	AT1G73590.1	Auxin efflux carrier family protein (PIN1)	1.287%
	1	28970875	AT1G77110.1	Auxin efflux carrier family protein (PIN1)	1.271%
	5	21055662	AT5G51810.1	Gibberellin 20 oxidase 2 (GA20OX2)	1.122%
Leaf development related	1	453280	AT1G02280.1	Translocon at the outer envelope membrane of chloroplasts 33 (TOC33)	1.574%
	1	1245256	AT1G04550.1	AUX/IAA transcriptional regulator family protein (AXR3)	1.725%
	1	1610455	AT1G05470.1	DNAse I-like superfamily protein	1.625%
	1	1682406	AT1G05630.1	Endonuclease/exonuclease/phosphatase family protein (Eepd1)	1.351%
Leaf morphology correlation	1	1105105	AT1G04180.1	YUCCA 9	1.381%

## Data Availability

Not applicable.

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
