# Peer review of "Functional Mapping of Genes Modulating Plant Shade Avoidance Using Leaf Traits"

_plants, 2023, doi:10.3390/plants12030608_

Round 1
Reviewer 1 Report
This manuscript entitled “Functional mapping of genes modulating plant shade avoidance using leaf traits” reported QTL mapping for 7/8 time points of leaf length, leaf area and petiole length by using 84 lines in both low and high density. In this study, shade avoidance phenotype was investigated as the allometric scaling of L-A and P-L during leaf ontogeny. This work may provide some new hints toward understanding regulation of leaf development and shade avoidance of Arabidopsis.
My major concern is these QTL mapping results were not confirmed in other population or other data such as gene expression or pathway analysis. The QTL mapping method for biparental mapping population should be used instead of GWAS method.
My minor concerns are as follows:
1/ In material and method part, the genotyping method and genetic linkage map information should be described in detail. Also, the QTL mapping method or software information should be provided.
2/ Legend of Figure 4 is incomplete.
3/ In Figure 5 and 6, it is very strange why the Y-axe was depicted as opposite wards? It may confuse readers.
4/ In addition, it appeared that sequence of Figure 6 and Figure 7 was wrong?
5/ What’s the difference between leaf length between in Figure 5A and 5C. Figure 5C-D were not cited in the text.
6/ It seemed that Figure7B was also not cited in the txt.
7/ Finally, a conclusion should be given. Actually, which SNPs were associated with shade avoidance confirmed in other population or other methods.
Author Response
My major concern is these QTL mapping results were not confirmed in other population or other data such as gene expression or pathway analysis. The QTL mapping method for biparental mapping population should be used instead of GWAS method.
Our response: Thank this reviewer for his/her enormous time and effort given to review our manuscript. S/he provided an unusually massive amount of reviews which we feel very useful for revising our manuscript and improving its presentation.
Let’s start with the first issue on ‘QTL result confirmation’. The reviewer exactly pointed out the drawback of this study, we believe if we have a second mapping population, e.g., the natural population of Arabidopsis, or the F2 population, QTLs mapped using this statistical method will be more convincing. However, based on the current research material, we hold the opinion that RIL population still has its own merits. Although one population, 40 replicates for each of the 84 progenies were used, this has allowed the design of high-density and low-density plantation, and allowed the subsequent comparison between these two settings. If F2 or natural population, a large number of progeny should be requested, since genotypes at most locus were not homozygous, and replicates requiring the same genotypes were not easy to gain. But this should become our suggestion to improve the further exploration of plant SAS. Also our point for this study is the genetic dissection of SAS through the allometric modeling approach. For the future research, we hope to construct a second mapping population for the more refined elaboration of QTL mapping results. The validation of gene expression or pathway analysis will also be our emphasis for the next step. For this concern reviewer#1 proposed, we had supplemented some sentence in discussion, see line 413-415 on page 15.
Second, for ‘biparental mapping population’, the reviewer exactly captured the feature of our QTL mapping using a biparent-derived population. Because of the ultra-high density of SNP marker for the whole Arabidopsis genome, and the association analysis was conducted on each SNP for QTL mapping, thus we habitually call it GWAS. Thanks for this suggestive advice, to avoid ambiguity but to maintain the association analysis using large amount of SNPs, we changed the expression as ‘the association analysis for each SNP’, see line 411 on page 15.
1/ In material and method part, the genotyping method and genetic linkage map information should be described in detail. Also, the QTL mapping method or software information should be provided.
Our response: Thanks to this reviewer for pointing out this issue, which has motivated us to provide details in calling SNPs. The genetic linkage map was not constructed, since the model plant Arabidopsis has a very-high-quality genome for reference, and SNP density is quite large, which helps the positioning of SNP in genome. The methodological detail for SNP genotyping was provided at line 111-127 on page 3. Also, for the QTL mapping method, we upload our R code with the Supplemental files.
2/ Legend of Figure 4 is incomplete.
Our response: It has been completed, see line 314-316 at page 9.
3/ In Figure 5 and 6, it is very strange why the Y-axe was depicted as opposite wards? It may confuse readers.
Our response: We opposite the Y-axe to compare the upper and lower parts. For convenience and to add the readability, we had follow this reviewer’s suggestion, see the new Figure 5 and Figure 7.
4/ In addition, it appeared that sequence of Figure 6 and Figure 7 was wrong?
Our response:We have corrected the sequence of Figure 6 and 7.
5/ What’s the difference between leaf length between in Figure 5A and 5C. Figure 5C-D were not cited in the text.
Our response: Figure 5A and 5C denote different genes of AT4G01895.1 and AT3G22170.1 respectively. With QTLs at different genes, the developmental trajectories governing leaf length for the two pair of allometry were different.
And we have cited 5C and 5D at line 351 and line 356 of page 12 to the text.
6/ It seemed that Figure7B was also not cited in the txt.
Our response: We have cited 7B to the text, see line 385 of page 13.
7/ Finally, a conclusion should be given. Actually, which SNPs were associated with shade avoidance confirmed in other population or other methods.
Our response: Thanks for reviewer#1’s reminder, this was quite the same question as the 1st question S/he proposed. For conclusion, we additionally add this section, see line 475-486 of page 16.
Reviewer 2 Report
The concept of the study is quite interesting and innovative, yet some clarifications/suggestions/modifications are suggested for your consideration and incorporation.
Title: What Do you mean by functional mapping of genes? Functional mapping needs to be explained in a better way. How QTL mapping resulted in gene mapping without any validation studies?
Line: 19-20: This should be 'recombinant inbred lines (RILs)' that were used to investigate the
developmental traits of Arabidopsis leaves....
Lines 29-30: The sentence is not very clear.
Lines 83-89: Sentences need re-writing for better understanding.
Lines 95-102: Need to be written in a more comprehensive way. currently, it looks more like a conclusion than an introduction.
Line 126: Change it to "At the time of taking the photograph,
...."
Line 195: Change this to 'for both the leaves'
Fig 4. What is LR on the Y-axis? Add the LOD score of the SNPs in the Figure.
Line 107: You mean 84 RILs? If so, this seems too less number for QTL mapping. What is the RIL size available with you? Add these details.
Line 131: Only details of the model for QTL mapping seem not enough. Which method/ software/ program was used for the QTL identification? What threshold LOD was decided for QTL identification? Mention all the desired details clearly.
In results: What is the PVE of the identified QTLs? How do you reach to the specific genes from any QTL region? These details are needed.
Line 387: What was QTL size and how many genes were there in the identified QTLs? How the mentioned genes are identified? These need through clarity, starting from the materials and methods, to the results and then also in the discussion section.
Overall the language of the manuscript is not up to the mark and needs editing. A few errors are pointed out in the manuscript.

Author Response
Let’s start with the titile. The reviewer said,” What Do you mean by functional mapping of genes? Functional mapping needs to be explained in a better way. How QTL mapping resulted in gene mapping without any validation studies? “
Our response: Functional mapping is a new QTL mapping method that Wu et al developed at 2002 (Genetics, 161,4:1751-1762). Specially, it uses dynamic or the developmental traits to map QTLs. For the dynamic traits, it integrated growth equations to describe the process of trait development. Thus, the kinetic feature with the developmental process can be calculated and depicted in a more precise manner. For this decade, it has been further developed and employed to solve many important issues in field of complex trait, which has a very strong feature of mathematical modeling. Due to this feature, we still use the ‘functional mapping’ with the title yet.
This study focused on using a new mathematical modeling approach to map genes mediating SAS, and is not like methods before. For the reliability of genes that the methods detected, we explain them through functional annotation. For validation, like reviewer#1 said, we think using a second population to perform the same experiment is a very good choice, by which the overlapped genes occured at both populations will be much reliable, and this will be our future emphasis, especially for their functional validation through molecular approach. In current study, though only one population, however, the treatment of setting high- and low-desnty environment, and experiments using 40 replicates per progeny can also increase the QTL reliability to a certein degree.
The reviewer said,’’ Line: 19-20: This should be 'recombinant inbred lines (RILs)' that were used to investigate the developmental traits of Arabidopsis leaves.
Our response: Thanks for pointing out this. On line 20 and line 102, it is now been corrected as “recombinant inbred lines (RILs)”.
Lines 29-30: The sentence is not very clear.
Our response: It is now been corrected as ‘By annotation, immune pathway genes, photoreceptor genes and phytohormone genes were identified to be involved in the SAS response.’.
Lines 83-89: Sentences need re-writing for better understanding.
Our response: The rewritten sentences are organized as follow, see line 85-91 of page 2.
In utilizing biological scaling allometry principles into QTL mapping, meanwhile, preserving the dynamic information with leaf development, we state to use functional mapping[13,14] to map SAS QTLs from the view of allometry QTL. Compared with other QTL mapping methods that using the static phenotype from only one timepoint, functional mapping using traits investigated at a series of timepoints, which can increase the model power for QTL detection, because the developmental process can well be modelled using mathematical equations with functional mapping.
Lines 95-102: Need to be written in a more comprehensive way. currently, it looks more like a conclusion than an introduction.
Our response: The rewritting sentences are organized as follow, see line 98-106 of page 3.
According to the advantage of functional mapping, we design two density conditions and incorporate the dynamic petiole and leaf traits from two density sets into functional mapping framework. From a methodological view, this strongly support the detection of SAS QTL but through a QTL density interaction way. Taking an Arabidopsis recombinant inbred line (RIL) as mapping population, we propose to use a bivariate functional mapping to study the genetic architecture of SAS in this study. Specially, how genes modulate the dynamic covary of petiole with leaf length, how leaf length covary with leaf area, and how genes mediate the interaction of leaf development and density environment, all these were synthesized in this study.
Line 126: Change it to "At the time of taking the photograph, ...."
Our response: Thanks. It is been changed.
Line 195: Change this to 'for both the leaves'
Our response: It is been changed.
Fig 4. What is LR on the Y-axis? Add the LOD score of the SNPs in the Figure.
Our response: LR is log-likelihood ratio (LR) test statistic under H0 and H1 hypothesis, and is calculated by LR = -2ln[L0(Y)/L1(Y)]. And LOD is a test statistic expressed as LOD = -2log10[L0(Y)/L1(Y)], which is actually equivalent to LR. According to the actual practice, we find we calculated LOD actually, now we have changed LR to LOD in Manhattan plot.
Line 107: You mean 84 RILs? If so, this seems too less number for QTL mapping. What is the RIL size available with you? Add these details.
Our response: Thanks for this pertinent suggestion. Yes, we admit 84 is a too less number for QTL mapping, also we realized the importance of using replicates to improve the heritability for QTL mapping. Actually, before this study, we performed simulation studies against the issue of small sample of progeny, this had guided us to set the 20 replicates to perform such experiment. With this number of replicates, and with the uniform environment to cultivate Arabidopsis, the QTL detection power can be improved. We have added details of replicates here, see line 111-112 of page 3.
Line 131: Only details of the model for QTL mapping seem not enough. Which method/ software/ program was used for the QTL identification? What threshold LOD was decided for QTL identification? Mention all the desired details clearly.
Our response: Thanks, just as reviewer#1 said, for the QTL mapping method, we upload our self-customized R code as supplementary files. For the threshold LOD, we use 1000 permutation test to determine the its value at a significance of 0.01, this information had been explained at line 207-211 of page 5.
In results: What is the PVE of the identified QTLs? How do you reach to the specific genes from any QTL region? These details are needed.
Our response: Thanks for this suggestion. For PVE, we now provide their value, which were calculated as the proportion of genetic variance against the total phenotypic variance. Because we analyze the allometry of L-A and P-L, which focused on the covary relationship of one trait with the other trait, thus the PVE will have a dynamic pattern. In order to provide a reliable result of PVE, when calculating PVE of L and P, we select a fixed value of A and L, according to the pair of L-A and P-L. For key SNPs from Table 1 and Table 2, the PVEs were now newly provided, see the last column in Table 1 and Table 2. Also, sentences indicating the result of PVE were also added, see line 310 and line 346.
For specific genes in QTL region, because the SNP density is quite large, and Arabidopsis has released a very-good quality genome publicly, through SNP position, we can reach their specific genes. Also see our response to the next question.
Line 387: What was QTL size and how many genes were there in the identified QTLs? How the mentioned genes are identified? These need through clarity, starting from the materials and methods, to the results and then also in the discussion section.
Our response: For QTL size, we had calculated their additive effects, since genotypes of QTLs in RIL population were segregates as QQ and qq (two homozygous genotypes), this can be seen at Figure 6. For shapeQTLs and positionQTLs, we respectively detected 545 and 418 significant SNPs. Yet, through functional annotation according to SNP position at genome level, 30 and 24 were located at protein genes, which had showed their functions in regulating SAS. All these had already been stated in origional manuscript. For clarification, with the identification of genes/QTLs found in our results, in meterials and methods, we added details of functional annotation, see section 2.5 at line 207-211 of page 5

Round 2
Reviewer 1 Report
After careful reading, the revised version did not satisfy me. My major concern is still the QTL mapping method. GWAS indeed can reveal some associated SNPs but can not give the linkage information and the interval QTL regions. I suggest that the authors perform genetic linkage analysis which may exclude unlinked SNPs. Also, the confirmation of the mapping result should be conducted, for instance in combination with pathway analysis, instead of choose a few genes (SNPs) in the text.
The English also needs further revision.

Author Response
After careful reading, the revised version did not satisfy me. My major concern is still the QTL mapping method. GWAS indeed can reveal some associated SNPs but can not give the linkage information and the interval QTL regions. I suggest that the authors perform genetic linkage analysis which may exclude unlinked SNPs. Also, the confirmation of the mapping result should be conducted, for instance in combination with pathway analysis, instead of choose a few genes (SNPs) in the text.
Our response: Thank you for your comment. We fully understand your concern, which is reasonable. Complex genetics include two approaches, one being linkage mapping based on a controlled cross, initiated with two contrasting parents, and the other being association mapping (or studies) based on a panel of samples drawn from natural populations. These two approaches have their own advantages and disadvantages: association mapping can make use of a wide spectrum of allelic variants, whereas linkage mapping detects some rare-allele loci. We chose linkage mapping given that growing evidence shows the importance of rare allele loci in trait control.
Linkage mapping is generally based on interval mapping, i.e., one can detect only one QTL on particular marker-marker intervals using the EM algorithm. This analytical approach has been approved to be powerful for sparse linkage maps. When the linkage map is high dense, a simple marker-phenotype association analysis is sufficient. We just use this kind of association analysis.
In this study, we did not report the linkage map, but our resequencing data contains information about the order of markers and their chromosomal distribution. Also, physical distances are broadly in agreement with genetic distances.
As to the question of “genetic linkage analysis which may exclude unlinked SNPs”, we address it in two ways. First, those unlinked SNPs, although failing to locate them to a linkage group due to imperfect sequencing arrays, may be still important determinants of complex traits. Excluding them may loss information. Second, all SNPs have actually been located to specific chromosomal regions in our study, suggesting no existence of unlinked SNPs. Third, we are very cautious about interpreting significant SNPs that come from a narrow region of the genome in an attempt to avoid the significance is due to linked SNPs rather than casual SNPs.
Based on these considerations, we feel that our analysis was in a good shape. We hope you agree with us, but we are very open to any further questions you may have on our manuscript. We are grateful for your time and effort devoted to review our work.
For pathway analysis, we now have performed a gene-enrichment analysis based on the annotated proteins with the reported number of significant SNPs. All these pathways can pave the way for the follow-up analysis of key genes related to immune pathway, photosensory genes, and phytohormone gene pathways. This section is arranged before section 3.5. See line 299-313 at page 9.
For sentences that marked with yellow color, we have now carefully revised to improve their expression, now marked with blue color.

Reviewer 2 Report
Suggested corrections have been incorporated.
Author Response
Thanks for the positive comments.
Round 3
Reviewer 1 Report
The authors have exactly answered my major concerns although further confirmation is needed. The English has also improved. I recommend it to be published on Plants.